# Can systematic skewness factors predict future interest rates: Evidence from China

**Xinyao Liang**[1]*, **Yichen Sun**[2]

**1** Business School, Chengdu University of Technology, Chengdu, Sichuan, China, **2** Business School, University of Bristol, Bristol, United Kingdom

* liangxinyao0009@qq.com

## Abstract

This study uses the numerical changes of the systematic skewness factors to reflect investors' preferences for the systematic skewness of stocks. As systematic skewness describes the correlation between individual stock returns and market volatility, stocks with positive systematic skewness can obtain positive returns during periods of market volatility. Thus, investors' preferences for the systematic skewness of stocks can reflect their hedging demands. By examining the predictive power of systematic skewness factors on future interest rates, we found that systematic skewness factors have a significant predictive power on future interest rates. This indicates that investors' hedging demands influence adjustments to interest rates by China's monetary authorities. Moreover, for both short-term and long-term interest rates, prediction errors based on systematic skewness factors are consistently lower than those from an AR model and the extended Taylor-rule model proposed by Ma et al. (2025). Systematic skewness can serve as an asymmetric pricing signal in the market for extreme interest rate risks. Its increase often indicates a rise in investors' anxiety over liquidity tightening, thereby providing central banks with a forward-looking sentiment monitoring window independent of traditional economic indicators.

## 1. Introduction

The incorporation of skewness into financial asset pricing models has garnered substantial academic attention, with a growing body of literature validating its empirical significance [1–3]. Given that skewness encapsulates multiple dimensions of information, a critical research focus in asset pricing involves decomposing skewness into systematic skewness, idiosyncratic skewness, and total skewness, followed by an independent examination of their respective pricing effects and underlying mechanisms. A growing body of research has consistently demonstrated that systematic skewness exhibits significant pricing effects in equity markets, whereas idiosyncratic skewness shows no statistically or economically meaningful pricing power [4–6]. This body of research collectively highlights the critical role of systematic skewness in

**Data availability statement:** All relevant data are within the paper and its Supporting information files.

**Funding:** This study is supported by Xinyao Liang's grant from the Natural Science Foundation of Sichuan Province [No. 2025NSFSC1959]. The funder had no role in study design, data collection and analysis, decision to publish, or preparation of the manuscript. There was no additional external funding received for this study.

**Competing interests:** The authors have declared that no competing interests exist.

both theoretical and empirical asset pricing, reinforcing its value as a key determinant of expected returns.

The existing literature on the formation mechanism of systematic skewness pricing effects has predominantly adopted an investor behavior perspective. Systematic skewness captures the co-movement between individual stock returns and market volatility. Stocks exhibiting positive systematic skewness demonstrate the ability to generate positive returns during periods of market turbulence, thereby satisfying investors' hedging demands. Langlois (2020) posits that risk-averse equity investors tend to overweight stocks with higher systematic skewness as a hedge against market volatility, resulting in the overvaluation of these securities and consequently lower future returns [2]. While a series of studies collectively establish a robust theoretical foundation for understanding skewness-based pricing effects, the current literature remains narrowly focused on investor behavior as the primary explanatory channel [7–10]. Notably, there appears to be a significant gap in the examination of macroeconomic determinants of systematic skewness pricing effects, as few studies to date have investigated this relationship through the lens of macroeconomic variables.

Our study follows the approach of Langlois (2020) and Harvey and Siddique (2000) to construct systematic skewness factors [2,11]. As defined in their work, systematic skewness reflects the correlation between individual stock returns and market volatility. Stocks with positive systematic skewness can provide investors with hedging protection during periods of high market volatility. Therefore, investors' preference for such stocks can be viewed as a proxy for their risk hedging demands. Building on this foundation, we examine the predictive power of the systematic skewness factors for future interest rates. This offers a macro-level explanation from the perspective of interest rate expectations for why the systematic skewness factors possess significant pricing effects, thereby addressing a gap in the existing literature regarding the role of macroeconomic factors.

We use China's overnight Shanghai Interbank Offered Rate (Shibor) and 10-year Treasury bond yield as proxies for short-term and long-term interest rates, respectively. The main conclusions of our study are as follows: The systematic skewness factors exhibit significant predictive power for future China's overnight Shibor and 10-year Treasury bond yield, with a negative correlation coefficient. This indicates that investors' current hedging demands affect future interest rates by influencing future monetary policy formulation. When investors' hedging demands rise, they increase purchases of stocks with positive systematic skewness, leading to the overvaluation of these stocks, lower future returns, and an increase in the current value of the systematic skewness factors. This signal is transmitted through monetary policy and significantly impacts future interest rates.

Building on this foundation, we conducted robustness tests through endogeneity tests, subsample tests, out-of-sample forecasting and comparisons with benchmark models. The results indicate that the systematic skewness factors indeed possess significant predictive power on future interest rates. Furthermore, whether predicting short-term or long-term interest rates, the prediction errors based on the systematic skewness factors are consistently smaller than those based on the AR model and the

extended Taylor rule model proposed by Ma et al. (2025) [12]. Meanwhile, our heterogeneity analysis reveals that in bull market environments, the systematic skewness factors exhibit stronger predictive power for short-term interest rates over its forecasting horizon compared to long-term interest rates. In contrast, during bear periods, the systematic skewness factors demonstrate a more prolonged predictive power for long-term interest rates than for short-term rates. This heterogeneity can be attributed to differences in investor behavior across varying market cycles.

Our empirical design and findings contribute to the existing research mainly in the following aspects: (1) From the perspective of investors' hedging demands on predicting future interest rates, it provides a macroeconomic explanation for the formation of the systematic skewness pricing effect. (2) It incorporates higher-moment risk pricing into the interest rate analysis framework, reveals the asymmetric pricing mechanism of the market toward extreme interest rate movements, and enriches the nonlinear theory of the interest rate term structure. (3) It offers new evidence on monetary policy transmission.

The structure of the remaining content of this paper is as follows. Section 2 outlines the definition and calculation method of systematic skewness. Section 3 provides a theoretical derivation of the transmission mechanism through which the systematic skewness factors predict future interest rates. Section 4 begins with an introduction to the data and variables used in this study, followed by an examination of the predictive power of the systematic skewness factors on future interest rates. Section 5 conducts robustness tests through endogeneity tests, subsample tests, out-of-sample forecasting and comparisons with benchmark models. Section 6 presents the main conclusions of our study.

## 2. Definition and calculation of systematic skewness

We use the numerical changes of systematic skewness factors to reflect the time-varying nature of investors' hedging demands. Therefore, we first construct appropriate systematic skewness to lay the foundation for testing the predictive power of systematic skewness factors on future interest rates in the subsequent analysis. Following the theoretical frameworks established by Langlois (2020) and Harvey and Siddique (2000) [2,11], systematic skewness captures the co-movement between individual stock returns and aggregate market volatility. The extant literature has developed three primary methodologies for measuring this relationship, whose computational procedures we formalize below.

**Methodology 1:** Building upon the classical capital asset pricing model (CAPM), we derive the three-moment CAPM specification as follows:

$$E_{t-1}\left[r_{i,t} - r_{f,t}\right] = \gamma_{M,t}Cov_{t-1}\left(r_{i,t}, r_{M,t}\right) + \gamma_{M^2,t}Cov_{t-1}\left(r_{i,t}, r_{M,t}^2\right) \tag{1}$$

where $r_{i,t}$ denotes the return of individual stock $i$ at time $t$, $r_{f,t}$ represents the risk-free rate, and $r_{M,t}$ is the market portfolio return. $\gamma_{M,t}$ and $\gamma_{M^2,t}$ signify the time-varying risk prices for covariance and coskewness, respectively. Additionally, $E_{t-1}[\cdot]$ denotes the conditional expectation operator based on information available at $t-1$, $Cov_{t-1}(\cdot)$ represents the conditional covariance operator. The coefficient in the quadratic term, $Cov_{t-1}\left(r_{i,t}, r_{M,t}^2\right)$ captures the sensitivity of individual stock returns to market volatility. We therefore define this as our first measure of systematic skewness, which we denote as *Cos*:

$$Cos_{i,t} = Cov_{t-1}\left(r_{i,t}, r_{M,t}^2\right) \tag{2}$$

**Methodology 2:** Following the theoretical framework of the three-moment CAPM, we employ an alternative regression-based approach to measure systematic skewness. Formally, we specify the regression model as follows:

$$E\left[r_{i,t} - r_{f,t}\right] = \beta_{M,i,t}\mu_{M,t} + \beta_{M^2,i,t}\mu_{M^2,t} \tag{3}$$

where $\mu_{M,t} = r_{M,t} - r_{f,t}$, $\mu_{M^2,t} = \mu_{M,t}^2$. Consequently, we establish the second operational definition of systematic skewness as the regression coefficient $\beta_{M^2,i,t}$, hereafter denoted as *BM2* for notational convenience.

**Methodology 3:** Harvey and Siddique (2000) developed a residual-based methodology for estimating systematic skewness [11]. Their approach proceeds in two stages:

First, market model residuals are obtained by estimating the standard CAPM specifications:

$$\epsilon_{i,t} = r_{i,t} - r_{f,t} - \alpha_i - \beta_{M,i}(r_{M,t} - r_{f,t}) \tag{4}$$

$$\epsilon_{M,t} = r_{M,t} - r_{i,t} - \mu_M \tag{5}$$

where $\mu_M = \frac{1}{T}\sum_{t=1}^{T}(r_{M,t} - r_{i,t})$ represents the historical average market excess return.

Subsequently, systematic skewness is quantified as the normalized conditional co-moment:

$$\beta_{HS,i,t} = \frac{E_{t-1}[\epsilon_{i,t}\epsilon_{M,t}^2]}{\sqrt{E_{t-1}[\epsilon_{i,t}^2]E_{t-1}[\epsilon_{M,t}^2]}} \tag{6}$$

This measure denoted $BHS$, captures the sensitivity of individual stock returns to squared market innovations, standardized by the volatility of both components.

## 3. Theoretical model derivation

We define the systematic skewness factors as the return difference between the 30% of stocks with the lowest (typically negative) systematic skewness and the 30% of stocks with the highest (typically positive) systematic skewness, and we employ numerical changes of systematic skewness factors to capture the time-varying nature of investors' hedging demands. For instance, when investors' hedging demands increase, they tend to favor stocks with positive systematic skewness, which drives up the prices of such stocks and leads to lower future returns, thereby resulting in an increase in the value of the systematic skewness factors. Conversely, when investors' hedging demands decrease, it will reduce the value of the systematic skewness factors. Building upon this foundation, we will examine the predictive power of the systematic skewness factors on future interest rates.

To further elaborate on the theory, we construct a two-period model to illustrate the relationship between investors' expectations of future interest rates and their hedging demands. We assume that investors have access to two types of assets – safe assets and risky assets, and that investors are risk-averse. When investors' hedging demand increases, they tend to sell risky assets and seek refuge in safe assets. Concurrently, we assume that the central bank is forward-looking and that its monetary policy rule follows some variant of the Taylor rule, meaning its interest rate decisions respond to expectations about future output gaps and inflation. According to Ma et al. (2025) [12], the reaction function of the Chinese monetary authorities must also account for exchange rate factors, in addition to the traditional Taylor rule framework. Therefore, we hypothesize that the reaction function of the Chinese monetary authorities follows the equation below:

$$r_{t+1}^* = r^* + E[\pi_{t+1}] + \beta_\pi(E(\pi_{t+1}) - \pi^*) + \beta_y(E(y_{t+1}) - y^*) + \beta_e(E(e_{t+1}) - e^*) + \varepsilon_{t+1} \tag{7}$$

where $r_{t+1}^*$ represents the desired future interest rate set by the Chinese monetary authorities based on current information, $r^*$ is the natural interest rate. $\pi_{t+1}$, $y_{t+1}$ and $e_{t+1}$ respectively denote the inflation level, economic output and exchange rate in period $t+1$, while $\pi^*$, $y^*$ and $e^*$ correspond to the central bank's inflation target, economic output target and exchange rate target, respectively. Moreover, $\beta_\pi$, $\beta_y$ and $\beta_e$ are all positive.

Since investors' hedging demands signal expectations of future economic recession, deflationary pressures and currency depreciation pressures [13–15], we specify the macroeconomic expectation equations as follows:

$$E(\pi_{t+1}) = f(\theta_t), \quad \frac{\partial f(\theta_t)}{\partial \theta_t} < 0 \tag{8}$$

$$E(y_{t+1}) = g(\theta_t), \quad \frac{\partial g(\theta_t)}{\partial \theta_t} < 0 \tag{9}$$

$$E(e_{t+1}) = h(\theta_t), \quad \frac{\partial h(\theta_t)}{\partial \theta_t} < 0 \tag{10}$$

where $\theta_t$ represents investors' hedging demands in period $t$. $f(\cdot)$, $g(\cdot)$, $h(\cdot)$ respectively represent the relationships between future expected inflation levels, economic output, exchange rates, and investors' current hedging demands.

Therefore, the reaction function of the Chinese monetary authorities should follow the equation below:

$$r_{t+1}^* = r^* + f(\theta_t) + \beta_\pi (f(\theta_t) - \pi^*) + \beta_y (g(\theta_t) - y^*) + \beta_e (h(\theta_t) - e^*) \tag{11}$$

Since $f(\theta_t)$, $g(\theta_t)$ and $h(\theta_t)$ are all decreasing functions of $\theta_t$, we can clearly observe that:

$$\frac{\partial r_{t+1}^*}{\partial \theta_t} = (1 + \beta_\pi) \frac{\partial f(\theta_t)}{\partial \theta_t} + \beta_y \frac{\partial g(\theta_t)}{\partial \theta_t} + \beta_e \frac{\partial h(\theta_t)}{\partial \theta_t} < 0 \tag{12}$$

From the above derivation, we can see that the future interest rate set by China's monetary authorities is a decreasing function of investors' current hedging demands. This implies that investors' current hedging demands affect future interest rates by influencing the formulation of future monetary policy. The relationship among hedging demands, systematic skewness factors and future interest rates is illustrated in the figure below Fig 1.

## 4. Empirical analysis

This section employs the overnight Shibor and 10-year Treasury bond yield as proxy variables for China's short-term and long-term interest rates, respectively. We subsequently examine the predictive power of systematic skewness factors for future interest rate movements across both time horizons.

### 4.1. Data and variables

To ensure sample quality and mitigate potential biases in our empirical analysis, we implement several standard data filters commonly adopted in Chinese market studies. Specifically, we exclude: (1) ST stocks, which are under special treatment due to financial distress; (2) stocks facing imminent delisting; (3) B-shares, which have different investor bases and regulatory regimes; (4) newly listed stocks (with less than 12 months of trading history) to avoid IPO effects; and

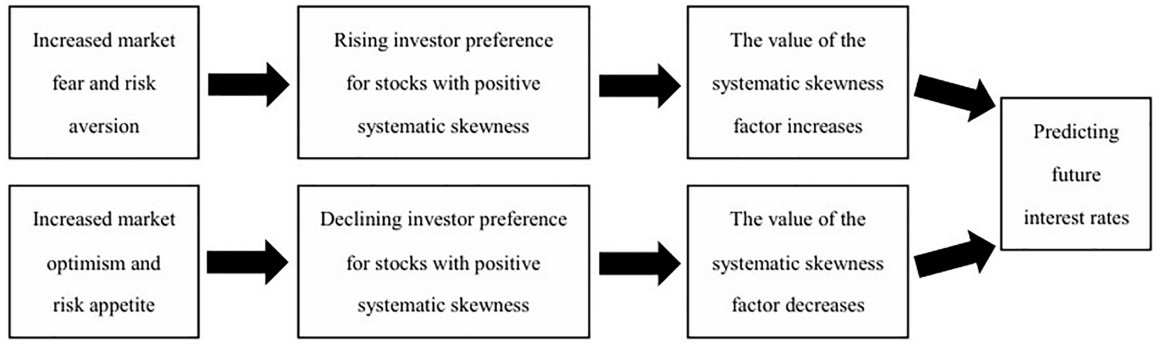

**Fig 1. The relationship among hedging demands, systematic skewness factors and future interest rates.**

(5) stocks with negative net assets. These filters help eliminate observations with extreme price volatility or fundamental weaknesses that could distort our factor pricing estimates.

After applying these screening criteria, our final sample comprises 4,980 individual stocks. Consistent with prior research on Chinese market microstructure, we restrict our analysis to the post-1996 period (January 1997 ~ December 2024) to maintain consistency in market regulations, as China implemented its current price limit system in December 1996, which may affect the distribution of stock price data. Table 1 presents the variables used in this study, along with the data frequency and sources required to calculate each variable.

Based on the grouping characteristic variable, we operationalize the computation of pricing factors as the monthly return spread between portfolios comprising the bottom 30% and top 30% of stocks, weighted by their respective market capitalizations. To illustrate, consider the systematic skewness factor construction: we first sort stocks into portfolios according to their systematic skewness values observed in month $t-1$, then calculate the factor return in month $t$ as the difference between the value-weighted returns of the lowest and highest 30% systematic skewness portfolios. This methodology is consistently applied to compute all classic pricing factors included as control variables, namely: the size factor (*SMB*), value factor (*HML*), momentum factor (*WML*), investment factor (*CMA*) and profitability factor (*RMW*). Each factor represents the return differential between the corresponding extreme portfolios sorted by their respective fundamental characteristics.

Due to data availability constraints during the factor construction process, our empirical analysis covers the period from January 2000 to December 2024, comprising 300 monthly observations. This sample period ensures consistent data coverage for both the systematic skewness factors and classic pricing factors. Table 2 reports the descriptive statistics of all pricing factors employed in this study.

## 4.2. Testing the predictive power of systematic skewness factors on future short-term interest rates

We begin by examining the predictive power of the systematic skewness factors on future short-term interest rates. Following the convention in monetary economics, we employ the overnight Shibor as our proxy for short-term interest rates. The baseline econometric specification is specified as follows:

$$Shibor_t = \alpha + \beta_1 SysFactor_{t-n} + \beta_2 X_t + \varepsilon_t \tag{13}$$

**Table 1. Variable name, data frequency and data source.**

| Variable Name | Data Frequency | Data Source |
|---|---|---|
| Individual Stock Turnover | Daily & Monthly | Wind<br>Bloomberg |
| Individual Stock Amplitude | | |
| Individual Stock Market Value | | |
| Shanghai Composite Index Amplitude | | |
| Shanghai Composite Index Turnover | | |
| Net Asset Return Rate | Annual | |
| Total Equity | | |
| Total Asset Growth Rate | | |
| GDP Annual Growth Rate | Quarterly | National Bureau of Statistics |
| M2 Annual Growth Rate | Monthly | People's Bank of China |
| Consumer Price Index (CPI) | | National Bureau of Statistics |
| US Effective Federal Funds Rate | | Federal Reserve Bank of St. Louis |
| Overnight Shibor | Daily | China Interbank Funding Center |
| Chinese 10-Year Treasury Bond Yields | | China Central Depository & Clearing Co., Ltd. |

**Table 2. Descriptive statistics of pricing factors.**

| Panel A | Descriptive Statistics of Systematic Skewness Factors | | | | | |
|---|---|---|---|---|---|---|
| Factor | Sample Size | Mean/% | Std. Dev./% | Median/% | Min/% | Max/% |
| *Cos* | 300 | 0.47 | 0.23 | 1.08 | −14.64 | 13.98 |
| *BM2* | 300 | 0.48 | 0.25 | 1.16 | −15.95 | 14.08 |
| *BHS* | 300 | 0.61 | 0.28 | 0.48 | −16.80 | 11.21 |
| Panel B | Descriptive Statistics of Classic Pricing Factors | | | | | |
| Factor | Sample Size | Mean/% | Std. Dev./% | Median/% | Min/% | Max/% |
| *SMB* | 300 | 0.82 | 0.37 | 1.25 | −16.02 | 21.33 |
| *HML* | 300 | −0.73 | 0.35 | −1.06 | −21.58 | 19.59 |
| *WML* | 300 | 0.28 | 0.22 | 0.38 | −18.41 | 16.53 |
| *CMA* | 300 | −0.37 | 0.34 | −0.41 | −17.49 | 13.88 |
| *RMW* | 300 | 0.54 | 0.43 | 0.62 | −17.49 | 13.88 |

Given that the overnight Shibor is reported at daily frequency, we compute the monthly average to align the interest rate data with our monthly factor observations. The vector $X_t$ comprises a comprehensive set of control variables, including both macroeconomic variables and classic pricing factors. Specifically, our macroeconomic variables consist of: (1) the annual growth rate of money supply (*M2*), (2) the Consumer Price Index (*CPI*), (3) the annual GDP growth rate (*GDP*), (4) the U.S. effective federal funds rate (*USR*) and (5) the annual growth rate of aggregate loans issued by financial institutions (*Loan*). Furthermore, we incorporate the following classic pricing factors as additional controls: (1) the size factor (*SMB*), (2) the value factor (*HML*), (3) the momentum factor (*WML*), (4) the investment factor (*CMA*) and (5) the profitability factor (*RMW*).

Table 3 presents the regression results examining the predictive relationship between systematic skewness factors and future overnight Shibor. The empirical evidence reveals several key findings: (1) We observe statistically significant predictive power for both the *BM2* and *BHS* factors in forecasting one- to three-month ahead overnight Shibor. The *Cos* factor demonstrates significant forecasting ability for shorter horizons of one to two months. (2) All three systematic skewness factors show statistically significant negative coefficients, supporting the hypothesis that increased hedging demands from investors will lead the Chinese monetary authorities to lower future interest rates. When investors' hedging demands increase, they tend to purchase more stocks with positive systematic skewness, leading to the overvaluation of such stocks, a decline in future returns, and an increase in the value of the systematic skewness factors. This signal is transmitted through monetary policy, significantly influencing future interest rates. This mechanism explains why the future overnight Shibor exhibits a negative correlation with the current systematic skewness factors.

### 4.3. Testing the predictive power of systematic skewness factors on future long-term interest rates

In Table 4, we use the 10-year Chinese Treasury bond yield (*YTB*) as a proxy for long-term interest rates and incorporate the control variables introduced earlier in Section 4.2 to conduct a full-sample regression. The baseline econometric specification is specified as follows:

$$YTB_t = \alpha + \beta_1 SysFactor_{t-n} + \beta_2 X_t + \varepsilon_t \tag{14}$$

The results show that the *Cos* and *BM2* factors exhibit significant predictive power for the 10-year Treasury yield over the next two months, while the *BHS* factor demonstrates significant predictive power on the 10-year Treasury yield over the next three months. Specifically, there is a significant negative correlation between the future 10-year yield and the systematic skewness factor. These findings indicate that when investors' hedging demands increase, monetary authorities

**Table 3. Predictive power of systematic skewness factors on short-term interest rates.**

$Shibor_t = \alpha + \beta_1 SysFactor_{t-n} + \beta_2 X_t + \varepsilon_t$

| $n=1$ | (1) | (2) | (3) | (4) | (5) | (6) | (7) | (8) | (9) |
|---|---|---|---|---|---|---|---|---|---|
| CONS | 0.027*** (20.72) | 0.022*** (9.52) | 0.022*** (7.93) | 0.027*** (20.57) | 0.022*** (9.42) | 0.021*** (7.86) | 0.027*** (20.24) | 0.022*** (9.39) | 0.021*** (7.81) |
| $\beta_{Cos}$ | −0.042** (−2.38) | −0.034** (−2.13) | −0.030** (−2.06) | | | | | | |
| $\beta_{BM2}$ | | | | −0.047*** (−2.61) | −0.039** (−2.40) | −0.036** (−2.32) | | | |
| $\beta_{BHS}$ | | | | | | | −0.052*** (−2.90) | −0.041*** (−2.67) | −0.038** (−2.59) |
| Macroeconomic Variables | NO | YES | YES | NO | YES | YES | NO | YES | YES |
| Classic Pricing Factors | NO | NO | YES | NO | NO | YES | NO | NO | YES |
| Adj.$R^2$ | 0.15 | 0.46 | 0.53 | 0.11 | 0.42 | 0.48 | 0.14 | 0.43 | 0.51 |
| $n=2$ | (1) | (2) | (3) | (4) | (5) | (6) | (7) | (8) | (9) |
| CONS | 0.027*** (20.73) | 0.022*** (9.61) | 0.022*** (8.01) | 0.027*** (20.59) | 0.022*** (9.50) | 0.021*** (7.97) | 0.027*** (20.29) | 0.022*** (9.45) | 0.021*** (7.89) |
| $\beta_{Cos}$ | −0.033* (−1.91) | −0.026* (−1.70) | −0.024* (−1.67) | | | | | | |
| $\beta_{BM2}$ | | | | −0.041** (−2.40) | −0.034** (−2.15) | −0.033** (−2.09) | | | |
| $\beta_{BHS}$ | | | | | | | −0.045*** (−2.67) | −0.036** (−2.46) | −0.035** (−2.38) |
| Macroeconomic Variables | NO | YES | YES | NO | YES | YES | NO | YES | YES |
| Classic Pricing Factors | NO | NO | YES | NO | NO | YES | NO | NO | YES |
| Adj.$R^2$ | 0.15 | 0.48 | 0.51 | 0.12 | 0.44 | 0.47 | 0.13 | 0.44 | 0.52 |
| $n=3$ | (1) | (2) | (3) | (4) | (5) | (6) | (7) | (8) | (9) |
| CONS | 0.027*** (20.69) | 0.022*** (9.55) | 0.021*** (8.03) | 0.027*** (20.61) | 0.022*** (9.54) | 0.021*** (7.90) | 0.027*** (20.38) | 0.022*** (9.49) | 0.021*** (7.68) |
| $\beta_{Cos}$ | −0.025 (−1.53) | −0.017 (−1.35) | −0.017 (−1.31) | | | | | | |
| $\beta_{BM2}$ | | | | −0.036** (−2.13) | −0.031* (−1.94) | −0.030* (−1.89) | | | |
| $\beta_{BHS}$ | | | | | | | −0.039** (−2.41) | −0.032** (−2.22) | −0.030** (−2.14) |
| Macroeconomic Variables | NO | YES | YES | NO | YES | YES | NO | YES | YES |
| Classic Pricing Factors | NO | NO | YES | NO | NO | YES | NO | NO | YES |
| Adj.$R^2$ | 0.14 | 0.45 | 0.50 | 0.11 | 0.40 | 0.47 | 0.13 | 0.42 | 0.51 |
| $n=4$ | (1) | (2) | (3) | (4) | (5) | (6) | (7) | (8) | (9) |
| CONS | 0.027*** (20.63) | 0.022*** (9.58) | 0.021*** (7.99) | 0.027*** (20.58) | 0.022*** (9.47) | 0.021*** (7.96) | 0.027*** (20.45) | 0.022*** (9.52) | 0.021*** (7.73) |
| $\beta_{Cos}$ | −0.011 (−1.06) | −0.06 (−0.70) | −0.05 (−0.67) | | | | | | |
| $\beta_{BM2}$ | | | | −0.022 (−1.56) | −0.19 (−1.30) | −0.14 (−1.15) | | | |
| $\beta_{BHS}$ | | | | | | | −0.021 (−1.48) | −0.13 (−1.09) | −0.10 (−0.87) |
| Macroeconomic Variables | NO | YES | YES | NO | YES | YES | NO | YES | YES |
| Classic Pricing Factors | NO | NO | YES | NO | NO | YES | NO | NO | YES |
| Adj.$R^2$ | 0.13 | 0.43 | 0.46 | 0.10 | 0.38 | 0.44 | 0.12 | 0.40 | 0.47 |

*Note:* The significance levels are adjusted using the Newey-West method, with the lag length selected as specified $T^{0.25} \approx 4$. ***, ** and * denote significance at the 1%, 5%, and 10% levels, respectively. The values in parentheses are Newey-West adjusted $t$ values.

## Table 4. Predictive power of systematic skewness factors on long-term interest rates.

$YTB_t = \alpha + \beta_1 SysFactor_{t-n} + \beta_2 X_t + \varepsilon_t$

| $n = 1$ | (1) | (2) | (3) | (4) | (5) | (6) | (7) | (8) | (9) |
|---|---|---|---|---|---|---|---|---|---|
| CONS | 0.036*** (30.23) | 0.025*** (14.63) | 0.025*** (12.75) | 0.036*** (30.19) | 0.025*** (14.65) | 0.025*** (12.66) | 0.036*** (29.77) | 0.025*** (14.57) | 0.025*** (12.67) |
| $\beta_{Cos}$ | -0.028** (-2.19) | -0.023** (-2.00) | -0.022** (-1.96) | | | | | | |
| $\beta_{BM2}$ | | | | -0.030** (-2.28) | -0.025** (-2.00) | -0.024** (-1.97) | | | |
| $\beta_{BHS}$ | | | | | | | -0.039*** (-2.92) | -0.033*** (-2.66) | -0.032** (-2.57) |
| Macroeconomic Variables | NO | YES | YES | NO | YES | YES | NO | YES | YES |
| Classic Pricing Factors | NO | NO | YES | NO | NO | YES | NO | NO | YES |
| Adj.$R^2$ | 0.11 | 0.38 | 0.50 | 0.09 | 0.37 | 0.47 | 0.13 | 0.41 | 0.51 |
| $n = 2$ | (1) | (2) | (3) | (4) | (5) | (6) | (7) | (8) | (9) |
| CONS | 0.036*** (30.20) | 0.025*** (14.48) | 0.025*** (12.59) | 0.036*** (30.19) | 0.025*** (14.57) | 0.025*** (12.75) | 0.036*** (29.79) | 0.025*** (14.58) | 0.025*** (12.60) |
| $\beta_{Cos}$ | -0.025* (-1.87) | -0.022* (-1.74) | -0.020* (-1.70) | | | | | | |
| $\beta_{BM2}$ | | | | -0.029* (-1.95) | -0.024* (-1.83) | -0.023* (-1.78) | | | |
| $\beta_{BHS}$ | | | | | | | -0.038** (-2.59) | -0.033** (-2.37) | -0.031** (-2.35) |
| Macroeconomic Variables | NO | YES | YES | NO | YES | YES | NO | YES | YES |
| Classic Pricing Factors | NO | NO | YES | NO | NO | YES | NO | NO | YES |
| Adj.$R^2$ | 0.10 | 0.38 | 0.49 | 0.09 | 0.36 | 0.47 | 0.12 | 0.41 | 0.50 |
| $n = 3$ | (1) | (2) | (3) | (4) | (5) | (6) | (7) | (8) | (9) |
| CONS | 0.036*** (30.22) | 0.025*** (14.53) | 0.025*** (12.70) | 0.036*** (30.23) | 0.025*** (14.58) | 0.025*** (12.69) | 0.036*** (29.80) | 0.025*** (14.57) | 0.025*** (12.28) |
| $\beta_{Cos}$ | -0.022 (-1.46) | -0.018 (-1.28) | -0.017 (-1.22) | | | | | | |
| $\beta_{BM2}$ | | | | -0.024 (-1.61) | -0.020 (-1.45) | -0.019 (-1.36) | | | |
| $\beta_{BHS}$ | | | | | | | -0.035** (-2.23) | -0.030** (-2.06) | -0.028* (-1.93) |
| Macroeconomic Variables | NO | YES | YES | NO | YES | YES | NO | YES | YES |
| Classic Pricing Factors | NO | NO | YES | NO | NO | YES | NO | NO | YES |
| Adj.$R^2$ | 0.10 | 0.37 | 0.51 | 0.09 | 0.38 | 0.47 | 0.10 | 0.41 | 0.49 |
| $n = 4$ | (1) | (2) | (3) | (4) | (5) | (6) | (7) | (8) | (9) |
| CONS | 0.036*** (30.15) | 0.025*** (14.60) | 0.025*** (12.68) | 0.036*** (30.19) | 0.025*** (14.52) | 0.025*** (12.74) | 0.036*** (29.86) | 0.025*** (14.52) | 0.025*** (12.23) |
| $\beta_{Cos}$ | -0.017 (-0.84) | -0.012 (-0.75) | -0.012 (-0.71) | | | | | | |
| $\beta_{BM2}$ | | | | -0.020 (-1.01) | -0.014 (-0.96) | -0.013 (-0.94) | | | |
| $\beta_{BHS}$ | | | | | | | -0.035 (-1.52) | -0.030 (-1.46) | -0.028 (-1.43) |
| Macroeconomic Variables | NO | YES | YES | NO | YES | YES | NO | YES | YES |
| Classic Pricing Factors | NO | NO | YES | NO | NO | YES | NO | NO | YES |
| Adj.$R^2$ | 0.09 | 0.35 | 0.49 | 0.08 | 0.36 | 0.45 | 0.08 | 0.39 | 0.48 |

*Note:* The significance levels are adjusted using the Newey-West method, with the lag length selected as specified $T^{0.25} \approx 4$. ***, ** and * denote significance at the 1%, 5%, and 10% levels, respectively. The values in parentheses are Newey-West adjusted $t$ values.

are likely to lower interest rates in the future to stabilize the market. This conclusion is consistent with the empirical evidence derived from the short-term interest rate analysis in Section 4.2.

## 5. Robustness tests

This section further examines the robustness of the empirical results through endogeneity tests, subsample tests, out-of-sample forecasting, comparisons with benchmark models, and predictive power tests based on alternative specifications for constructing systematic skewness factors.

### 5.1. Endogeneity tests

This section includes Granger causality tests and instrumental variable tests.

**5.1.1. Granger causality tests.** Table 5 presents the Granger causality tests between the systematic skewness factors and future interest rates. It can be observed that the *Cos*, *BM2*, and *BHS* factors are Granger causes of both future short-term and long-term interest rates, while future short-term and long-term interest rates are not Granger causes of the *Cos*, *BM2*, and *BHS* factors. The results indicate that the systematic skewness factors have a unidirectional impact on both future short-term and long-term interest rates.

**5.1.2. Instrumental variable tests.** We select the VIX index from the U.S. market as an instrumental variable for the systematic skewness factors. The VIX index reflects the future risk expectations of the core market in the global financial system, and its fluctuations quickly influence global market sentiment, including that of Chinese stock market, through channels such as capital flows, adjustments in multinational investor behavior, and contagion of risk preferences. As an external variable, the VIX index is not directly correlated with the internal fundamental factors of the Chinese stock market (such as domestic corporate earnings or policy changes), satisfying the exogeneity condition of an instrumental variable. Meanwhile, its systematic correlation with investor sentiment in the Chinese stock market ensures relevance.

Table 6 examines the predictive power of the VIX index on future interest rates. The results show that using the VIX index from the U.S. market as an instrumental variable for Chinese investor market sentiment reveals its significant predictive ability for short-term interest rates over the next five months and long-term interest rates over the next four months. This finding indicates that although the VIX index itself reflects volatility expectations in the U.S. stock market, it

**Table 5. Granger causality tests.**

| Panel A | Granger Causality Tests between Systematic Skewness Factors and Future Short-Term Interest Rates | | | |
|---|---|---|---|---|
| | **Null Hypothesis** | **Obs** | **F-Statistic** | **Probability** |
| (1) | *Cos* does not Granger Cause *Shibor* | 213 | 5.0030*** | 0.0075 |
| (2) | *Shibor* does not Granger Cause *Cos* | 213 | 0.6152 | 0.5415 |
| (3) | *BM2* does not Granger Cause *Shibor* | 213 | 5.1990*** | 0.0062 |
| (4) | *Shibor* does not Granger Cause *BM2* | 213 | 1.0914 | 0.3376 |
| (5) | *BHS* does not Granger Cause *Shibor* | 213 | 5.4801*** | 0.0048 |
| (6) | *Shibor* does not Granger Cause *BHS* | 213 | 0.9874 | 0.3742 |
| Panel B | Granger Causality Tests between Systematic Skewness Factors and Future Long-Term Interest Rates | | | |
| | Null Hypothesis | Obs | F-Statistic | Probability |
| (7) | *Cos* does not Granger Cause *YTB* | 257 | 3.0128* | 0.0509 |
| (8) | *YTB* does not Granger Cause *Cos* | 257 | 0.4260 | 0.6536 |
| (9) | *BM2* does not Granger Cause *YTB* | 257 | 3.5070** | 0.0314 |
| (10) | *YTB* does not Granger Cause *BM2* | 257 | 0.5490 | 0.5782 |
| (11) | *BHS* does not Granger Cause *YTB* | 257 | 3.8434** | 0.0227 |
| (12) | *YTB* does not Granger Cause *BHS* | 257 | 0.7543 | 0.4714 |

**Table 6. Instrumental variable tests.**

| Panel A | $Shibor_t = \alpha + \beta_1 VIX_{t-n} + \beta_2 X_t + \varepsilon_t$ | | | | | |
|---|---|---|---|---|---|---|
| | $n = 1$ | $n = 2$ | $n = 3$ | $n = 4$ | $n = 5$ | $n = 6$ |
| CONS | 0.028*** (11.90) | 0.027*** (11.43) | 0.026*** (11.39) | 0.025*** (10.55) | 0.024*** (10.14) | 0.023*** (9.78) |
| $\beta_{VIX}$ | $-3.50 \times 10^{-4***}$ (−4.98) | $-3.21 \times 10^{-4***}$ (−4.29) | $-3.26 \times 10^{-4***}$ (−3.76) | $-2.48 \times 10^{-4***}$ (−2.82) | $-2.01 \times 10^{-4**}$ (−1.98) | $-1.47 \times 10^{-4}$ (−1.15) |
| Macroeconomic Variables | YES | YES | YES | YES | YES | YES |
| Adj.$R^2$ | 0.43 | 0.41 | 0.40 | 0.37 | 0.35 | 0.32 |
| Panel B | $YTB_t = \alpha + \beta_1 VIX_{t-n} + \beta_2 X_t + \varepsilon_t$ | | | | | |
| | $n = 1$ | $n = 2$ | $n = 3$ | $n = 4$ | $n = 5$ | $n = 6$ |
| CONS | 0.035*** (19.96) | 0.035*** (20.05) | 0.035*** (20.36) | 0.035*** (20.79) | 0.036*** (21.35) | 0.037*** (22.16) |
| $\beta_{VIX}$ | $-1.34 \times 10^{-4***}$ (−2.86) | $-1.20 \times 10^{-4**}$ (−2.49) | $-1.17 \times 10^{-4**}$ (−2.25) | $-1.04 \times 10^{-4*}$ (−1.95) | $-0.75 \times 10^{-4}$ (−1.43) | $-0.80 \times 10^{-4}$ (−1.50) |
| Macroeconomic Variables | YES | YES | YES | YES | YES | YES |
| Adj.$R^2$ | 0.32 | 0.32 | 0.31 | 0.29 | 0.27 | 0.28 |

*Note:* The significance levels are adjusted using the Newey-West method, with the lag length selected as specified $T^{0.25} \approx 4$. ***, ** and * denote significance at the 1%, 5%, and 10% levels, respectively. The values in parentheses are Newey-West adjusted *t* values.

can influence fund flows driven by risk aversion sentiment and changes in policy expectations in Chinese stock market through the global financial sentiment transmission mechanism. Specifically, when the VIX index rises (indicating heightened global risk aversion), investors in the Chinese stock market may increase their purchases of stocks with positive systematic skewness due to elevated risk aversion, thereby affecting the value of the systematic skewness factors. When Chinese monetary authorities detect such market panic, they are likely to lower interest rates to maintain market stability. Conversely, a decline in the VIX index may drive interest rates upward. Therefore, a significant negative correlation exists between future interest rates and the current VIX index.

### 5.2. Subsample tests

Based on the modified BB turning point method [16], the Chinese stock market's bull and bear market cycles are divided into periods, as shown in Fig 2. We can find that from January 2000 to December 2024, the Chinese stock market can be divided into six bull and bear cycles, with a cumulative bull market duration of 132 months and a cumulative bear market duration of 168 months.

In the following analysis, we use the overnight Shibor and 10-year Treasury bond yield as proxy variables for China's short-term and long-term interest rates, respectively, and examine the predictive power of the systematic skewness factors on future interest rates.

**5.2.1. Empirical analysis based on bull markets.** The data processing and control variable selection in Table 7 follow the same procedure as in the full-sample regression. Regarding the prediction of short-term interest rates, the results indicate that during bull market periods, both the *BM2* and *BHS* factors exhibit significant predictive power for the overnight Shibor at the three-month horizon, while the *Cos* factor shows significant predictive ability at the two-month horizon. A significantly negative correlation is observed between future overnight Shibor and all three systematic skewness factors. For long-term interest rate prediction, the findings reveal that during bull markets, the *BM2* and *BHS* factors significantly predict the 10-year Treasury bond yield at the two-month horizon, whereas the *Cos* factor demonstrates predictive power at the one-month horizon. Similarly, a significantly negative correlation is identified between the 10-year Treasury bond yield and each of the three systematic skewness factors.

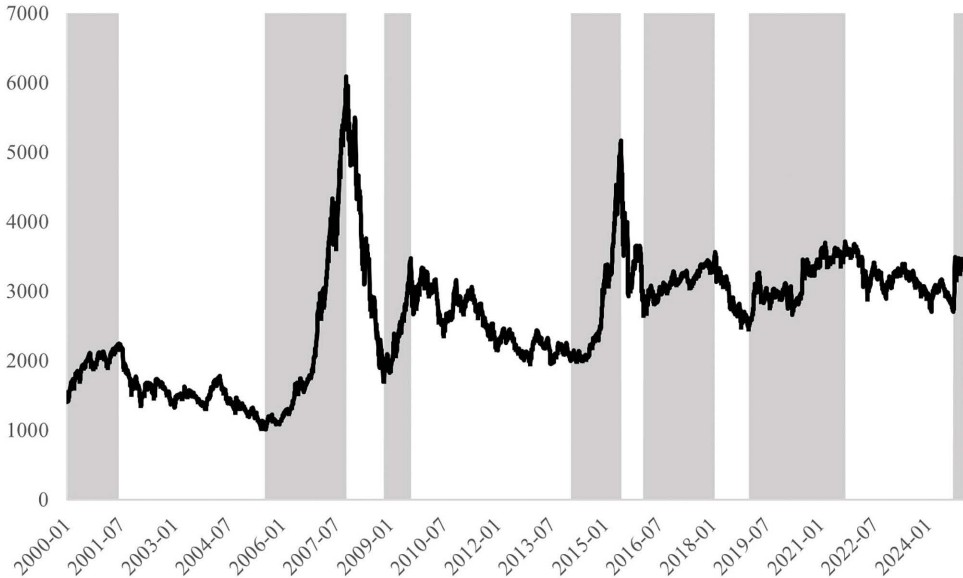

**Fig 2. The division of bull and bear markets in the Chinese stock market.** *Note:* Fig 2 shows the Shanghai Composite Index points from January 2000 to December 2024. The shaded areas represent bull market periods, while the unshaded areas represent bear market periods.

Our conclusions remain consistent with the empirical results obtained from the full-sample analysis. However, in bull market environments, the systematic skewness factors exhibit predictive power over a longer horizon on short-term interest rates than long-term interest rates. This pattern can be attributed to the fact that during bull markets, monetary authorities pay closer attention to market sentiment in adjusting short-term interest rates to maintain market stability [17], making short-term rates more sensitive to risk perceptions based on systematic skewness. In contrast, long-term interest rates are influenced more strongly by structural macroeconomic factors, which may attenuate the predictive advantage of systematic skewness measures over extended periods [18]. Additionally, the prevailing optimism and higher risk appetite in bullish phases may lead investors to employ shorter-term hedging strategies, further strengthening the persistent predictive power of systematic skewness on short-term rates [19,20].

**5.2.2. Empirical analysis based on bear markets.** Regarding the prediction of short-term interest rates, the regression results in Table 8 indicate that during bear market phases, both the *BM2* and *BHS* factors exhibit significant predictive power for the overnight Shibor at the two-month horizon, while the *Cos* factor also shows significant predictive ability at the one-month horizon. A significantly negative correlation is observed between the future overnight Shibor and all three systematic skewness factors. With respect to long-term interest rates, the results show that during bear markets, the *BM2* and *BHS* factors demonstrate significant predictive power for the 10-year Treasury bond yield at the three-month horizon, whereas the *Cos* factor exhibits predictive power at the two-month horizon. Similarly, a significantly negative correlation exists between the future 10-year Treasury bond yield and the three systematic skewness factors.

The systematic skewness factors exhibit predictive power over a longer horizon on long-term interest rates than short-term rates – a conclusion that contrasts with the findings observed in bull market conditions. This discrepancy may be attributed to the distinct behaviors and expectations of investors during bear markets. In periods of sustained decline, investors tend to increase their allocation to long-term safe-haven assets to hedge against the risk of falling asset prices. As a result, changes in long-term interest rates primarily reflect current market risk aversion and pessimistic expectations about future economic prospects [21]. In contrast, short-term rates are more susceptible to immediate monetary policies and frequent market fluctuations, resulting in a relatively shorter predictive window. Furthermore, the flight-to-safety

**Table 7. Predictive power of systematic skewness factors on future interest rates (bull markets).**

| $Shibor_t = \alpha + \beta_1 SysFactor_{t-n} + \beta_2 X_t + \varepsilon_t$ | | | | $YTB_t = \alpha + \beta_1 SysFactor_{t-n} + \beta_2 X_t + \varepsilon_t$ | | | |
|---|---|---|---|---|---|---|---|
| $n = 1$ | **(1)** | **(2)** | **(3)** | $n = 1$ | **(1)** | **(2)** | **(3)** |
| CONS | 0.030*** (11.76) | 0.027*** (11.65) | 0.027*** (11.36) | CONS | 0.046*** (18.78) | 0.035*** (18.37) | 0.035*** (18.19) |
| $\beta_{Cos}$ | −0.043** (−2.29) | | | $\beta_{Cos}$ | −0.027** (−2.03) | | |
| $\beta_{BM2}$ | | −0.065** (−2.56) | | $\beta_{BM2}$ | | −0.039*** (−2.18) | |
| $\beta_{BHS}$ | | | −0.078*** (−2.75) | $\beta_{BHS}$ | | | −0.043*** (−2.51) |
| Macroeconomic Variables | YES | YES | YES | Macroeconomic Variables | YES | YES | YES |
| $Adj.R^2$ | 0.59 | 0.60 | 0.62 | $Adj.R^2$ | 0.46 | 0.48 | 0.48 |
| $n = 2$ | (1) | (2) | (3) | $n = 2$ | (1) | (2) | (3) |
| CONS | 0.029*** (11.58) | 0.026*** (11.61) | 0.025*** (11.25) | CONS | 0.044*** (18.72) | 0.033*** (18.47) | 0.033*** (18.03) |
| $\beta_{Cos}$ | −0.039** (−1.98) | | | $\beta_{Cos}$ | −0.019 (−1.56) | | |
| $\beta_{BM2}$ | | −0.051** (−2.27) | | $\beta_{BM2}$ | | −0.031* (−1.75) | |
| $\beta_{BHS}$ | | | −0.062** (−2.53) | $\beta_{BHS}$ | | | −0.035* (−1.93) |
| Macroeconomic Variables | YES | YES | YES | Macroeconomic Variables | YES | YES | YES |
| $Adj.R^2$ | 0.61 | 0.57 | 0.60 | $Adj.R^2$ | 0.45 | 0.49 | 0.51 |
| $n = 3$ | (1) | (2) | (3) | $n = 3$ | (1) | (2) | (3) |
| CONS | 0.028*** (11.67) | 0.028*** (11.56) | 0.027*** (11.14) | CONS | 0.042*** (18.51) | 0.032*** (18.28) | 0.030*** (18.12) |
| $\beta_{Cos}$ | −0.026 (−1.02) | | | $\beta_{Cos}$ | −0.006 (−0.57) | | |
| $\beta_{BM2}$ | | −0.037* (−1.80) | | $\beta_{BM2}$ | | −0.021 (−1.43) | |
| $\beta_{BHS}$ | | | −0.043** (−2.02) | $\beta_{BHS}$ | | | −0.020 (−1.35) |
| Macroeconomic Variables | YES | YES | YES | Macroeconomic Variables | YES | YES | YES |
| $Adj.R^2$ | 0.59 | 0.55 | 0.62 | $Adj.R^2$ | 0.47 | 0.46 | 0.47 |
| $n = 4$ | (1) | (2) | (3) | $n = 4$ | (1) | (2) | (3) |
| CONS | 0.027*** (11.39) | 0.025*** (11.25) | 0.025*** (11.08) | CONS | 0.042*** (18.63) | 0.031*** (18.31) | 0.032*** (18.06) |
| $\beta_{Cos}$ | −0.009 (−0.41) | | | $\beta_{Cos}$ | −0.007 (−0.79) | | |
| $\beta_{BM2}$ | | −0.013 (−0.62) | | $\beta_{BM2}$ | | −0.015 (−1.28) | |
| $\beta_{BHS}$ | | | −0.015 (−0.73) | $\beta_{BHS}$ | | | −0.016 (−1.33) |
| Macroeconomic Variables | YES | YES | YES | Macroeconomic Variables | YES | YES | YES |
| $Adj.R^2$ | 0.57 | 0.56 | 0.59 | $Adj.R^2$ | 0.45 | 0.44 | 0.45 |

*Note:* The significance levels are adjusted using the Newey-West method, with the lag length selected as specified $T^{0.25} \approx 4$. ***, ** and * denote significance at the 1%, 5%, and 10% levels, respectively. The values in parentheses are Newey-West adjusted $t$ values.

# Table 8. Predictive power of systematic skewness factors on future interest rates (bear markets).

$Shibor_t = \alpha + \beta_1 SysFactor_{t-n} + \beta_2 X_t + \varepsilon_t$ | | | | $YTB_t = \alpha + \beta_1 SysFactor_{t-n} + \beta_2 X_t + \varepsilon_t$ | | | |
|---|---|---|---|---|---|---|---|
| $n = 1$ | **(1)** | **(2)** | **(3)** | $n = 1$ | **(1)** | **(2)** | **(3)** |
| CONS | 0.017*** (7.63) | 0.017*** (7.39) | 0.017*** (7.25) | CONS | 0.015*** (12.75) | 0.015*** (12.62) | 0.015*** (12.39) |
| $\beta_{Cos}$ | −0.014* (−1.78) | | | $\beta_{Cos}$ | −0.018** (−2.21) | | |
| $\beta_{BM2}$ | | −0.023** (−1.96) | | $\beta_{BM2}$ | | −0.027** (−2.57) | |
| $\beta_{BHS}$ | | | −0.029** (−2.19) | $\beta_{BHS}$ | | | −0.032*** (−2.62) |
| Macroeconomic Variables | YES | YES | YES | Macroeconomic Variables | YES | YES | YES |
| Adj.R² | 0.34 | 0.35 | 0.38 | Adj.R² | 0.35 | 0.37 | 0.40 |
| $n = 2$ | (1) | (2) | (3) | $n = 2$ | (1) | (2) | (3) |
| CONS | 0.017*** (7.58) | 0.017*** (7.28) | 0.017*** (7.12) | CONS | 0.015*** (12.85) | 0.015*** (12.58) | 0.015*** (12.45) |
| $\beta_{Cos}$ | −0.010 (−1.37) | | | $\beta_{Cos}$ | −0.016* (−1.93) | | |
| $\beta_{BM2}$ | | −0.021* (−1.75) | | $\beta_{BM2}$ | | −0.024** (−2.16) | |
| $\beta_{BHS}$ | | | −0.025* (−1.90) | $\beta_{BHS}$ | | | −0.028** (−2.24) |
| Macroeconomic Variables | YES | YES | YES | Macroeconomic Variables | YES | YES | YES |
| Adj.R² | 0.36 | 0.33 | 0.39 | Adj.R² | 0.36 | 0.36 | 0.38 |
| $n = 3$ | (1) | (2) | (3) | $n = 3$ | (1) | (2) | (3) |
| CONS | 0.017*** (7.45) | 0.017*** (7.42) | 0.016*** (7.25) | CONS | 0.015*** (12.95) | 0.015*** (12.60) | 0.015*** (12.40) |
| $\beta_{Cos}$ | −0.009 (−1.13) | | | $\beta_{Cos}$ | −0.013 (−1.42) | | |
| $\beta_{BM2}$ | | −0.017 (−1.34) | | $\beta_{BM2}$ | | −0.020* (−1.83) | |
| $\beta_{BHS}$ | | | −0.020 (−1.52) | $\beta_{BHS}$ | | | −0.021* (−1.95) |
| Macroeconomic Variables | YES | YES | YES | Macroeconomic Variables | YES | YES | YES |
| Adj.R² | 0.34 | 0.35 | 0.37 | Adj.R² | 0.34 | 0.39 | 0.40 |
| $n = 4$ | (1) | (2) | (3) | $n = 4$ | (1) | (2) | (3) |
| CONS | 0.017*** (7.60) | 0.017*** (7.55) | 0.016*** (7.15) | CONS | 0.015*** (12.82) | 0.014*** (12.42) | 0.014*** (12.25) |
| $\beta_{Cos}$ | −0.007 (−0.86) | | | $\beta_{Cos}$ | −0.009 (−0.95) | | |
| $\beta_{BM2}$ | | −0.012 (−0.91) | | $\beta_{BM2}$ | | −0.015 (−1.21) | |
| $\beta_{BHS}$ | | | −0.012 (−1.09) | $\beta_{BHS}$ | | | −0.018 (−1.36) |
| Macroeconomic Variables | YES | YES | YES | Macroeconomic Variables | YES | YES | YES |
| Adj.R² | 0.32 | 0.38 | 0.40 | Adj.R² | 0.36 | 0.37 | 0.39 |

*Note:* The significance levels are adjusted using the Newey-West method, with the lag length selected as specified $T^{0.25} \approx 4$. ***, ** and * denote significance at the 1%, 5%, and 10% levels, respectively. The values in parentheses are Newey-West adjusted $t$ values.

phenomenon tends to be more pronounced and structural during bear markets, amplifying the prolonged predictive relationship between skewness factors and long-term yields [22,23].

## 5.3. Out-of-sample forecasting and comparisons with benchmark models

This section fits the model using historical data from the past five years and forecasts monthly interest rates for the next twelve months. The prediction errors are measured using MSE, RMSE and MAE, with the results shown in Table 9. Whether predicting short-term or long-term interest rates, the prediction errors based on the systematic skewness factors are significantly smaller than those based on the AR model and the extended Taylor rule model proposed by Ma et al. (2025) [12].

## 5.4. Predictive power tests based on alternative specifications for constructing systematic skewness factors

In this section, the systematic skewness factors are redefined as the return difference between the stocks in the lowest 10% (typically negative) and the highest 10% (typically positive) of systematic skewness. The predictive power of these new systematic skewness factors on future interest rates is examined, and the results are shown in Table 10. For short-term interest rates, all the *Cos*, *BM2* and *BHS* factors have significant predictive power on overnight Shibor over the next 1–3 months. For long-term interest rates, the *BHS* factor has significant predictive power on the 10-year Treasury yield over the next 1–2 months, whereas the *Cos* and *BM2* factors only show significant predictive power on the 10-year Treasury yield over the next 1 month.

## 6. Conclusions

While existing research indicates that systematic skewness exerts a significant pricing impact on the stock market, the current literature remains narrowly focused on investor behavior as the primary explanatory channel for the origin of this effect. There is still insufficient in-depth exploration of how macroeconomic variables influence investors' preference for stocks characterized by systematic skewness. This study utilizes the fluctuations in the systematic skewness factors to capture the time-varying nature of investor hedging demands. By investigating the predictive power of the systematic skewness factors on future interest rates, we provide a macroeconomic explanation for the formation of the systematic skewness pricing effect. Concurrently, we incorporate higher-moment risk pricing into the interest rate analysis framework,

**Table 9. Out-of-sample forecasting and comparisons with AR model and extended Taylor rule model by Ma et al. (2025).**

| Panel A | Out-of-Sample Forecasting of Short-Term Interest Rates | | |
|---|---|---|---|
| Models | MSE | RMSE | MAE |
| Predicting Shibor Based on *Cos* Factor | $3.15 \times 10^{-6}$ | 0.0018 | 0.0015 |
| Predicting Shibor Based on *BM2* Factor | $2.89 \times 10^{-6}$ | 0.0017 | 0.0013 |
| Predicting Shibor Based on *BHS* Factor | $2.34 \times 10^{-6}$ | 0.0015 | 0.0011 |
| Predicting Shibor Based on AR Model | $4.88 \times 10^{-6}$ | 0.0028 | 0.0023 |
| Predicting Shibor Based on Extended Taylor Rule Model by Ma et al. (2025) | $7.45 \times 10^{-6}$ | 0.0047 | 0.0034 |
| Panel B | Out-of-Sample Forecasting of Long-Term Interest Rates | | |
| Models | MSE | RMSE | MAE |
| Predicting YTB Based on *Cos* Factor | $7.13 \times 10^{-6}$ | 0.0027 | 0.0023 |
| Predicting YTB Based on *BM2* Factor | $6.87 \times 10^{-6}$ | 0.0026 | 0.0022 |
| Predicting YTB Based on *BHS* Factor | $6.42 \times 10^{-6}$ | 0.0025 | 0.0021 |
| Predicting YTB Based on AR Model | $7.75 \times 10^{-6}$ | 0.0028 | 0.0025 |
| Predicting YTB Based on Extended Taylor Rule Model by Ma et al. (2025) | $10.72 \times 10^{-6}$ | 0.0041 | 0.0035 |

**Table 10. Predictive power tests based on alternative specifications for constructing systematic skewness factors.**

| $Shibor_t = \alpha + \beta_1 SysFactor_{t-n} + \beta_2 X_t + \varepsilon_t$ | | | | $YTB_t = \alpha + \beta_1 SysFactor_{t-n} + \beta_2 X_t + \varepsilon_t$ | | | |
|---|---|---|---|---|---|---|---|
| $n = 1$ | **(1)** | **(2)** | **(3)** | $n = 1$ | **(1)** | **(2)** | **(3)** |
| CONS | 0.028*** (8.57) | 0.027*** (8.36) | 0.027*** (8.13) | CONS | 0.033*** (12.90) | 0.033*** (12.34) | 0.032*** (12.29) |
| $\beta_{Cos}$ | −0.047** (−2.35) | | | $\beta_{Cos}$ | −0.038** (−2.08) | | |
| $\beta_{BM2}$ | | −0.055** (−2.50) | | $\beta_{BM2}$ | | −0.045** (−2.12) | |
| $\beta_{BHS}$ | | | −0.060*** (−2.72) | $\beta_{BHS}$ | | | −0.049** (−2.31) |
| Macroeconomic Variables | YES | YES | YES | Macroeconomic Variables | YES | YES | YES |
| $Adj.R^2$ | 0.47 | 0.50 | 0.52 | $Adj.R^2$ | 0.43 | 0.45 | 0.46 |
| $n = 2$ | (1) | (2) | (3) | $n = 2$ | (1) | (2) | (3) |
| CONS | 0.028*** (8.66) | 0.028*** (8.49) | 0.027*** (8.40) | CONS | 0.033*** (12.86) | 0.032*** (12.35) | 0.032*** (12.15) |
| $\beta_{Cos}$ | −0.041** (−2.07) | | | $\beta_{Cos}$ | −0.029 (−1.58) | | |
| $\beta_{BM2}$ | | −0.048** (−2.17) | | $\beta_{BM2}$ | | −0.036 (−1.62) | |
| $\beta_{BHS}$ | | | −0.052** (−2.43) | $\beta_{BHS}$ | | | −0.038* (−1.83) |
| Macroeconomic Variables | YES | YES | YES | Macroeconomic Variables | YES | YES | YES |
| $Adj.R^2$ | 0.45 | 0.51 | 0.51 | $Adj.R^2$ | 0.41 | 0.44 | 0.44 |
| $n = 3$ | (1) | (2) | (3) | $n = 3$ | (1) | (2) | (3) |
| CONS | 0.028*** (8.73) | 0.028*** (8.46) | 0.027*** (8.29) | CONS | 0.033*** (12.66) | 0.032*** (12.25) | 0.032*** (12.14) |
| $\beta_{Cos}$ | −0.036* (−1.78) | | | $\beta_{Cos}$ | −0.013 (−0.67) | | |
| $\beta_{BM2}$ | | −0.040* (−1.80) | | $\beta_{BM2}$ | | −0.017 (−0.73) | |
| $\beta_{BHS}$ | | | −0.043** (−2.02) | $\beta_{BHS}$ | | | −0.019 (−0.95) |
| Macroeconomic Variables | YES | YES | YES | Macroeconomic Variables | YES | YES | YES |
| $Adj.R^2$ | 0.46 | 0.49 | 0.52 | $Adj.R^2$ | 0.41 | 0.43 | 0.45 |
| $n = 4$ | (1) | (2) | (3) | $n = 4$ | (1) | (2) | (3) |
| CONS | 0.027*** (8.44) | 0.027*** (8.19) | 0.027*** (8.15) | CONS | 0.032*** (12.79) | 0.032*** (12.21) | 0.032*** (12.16) |
| $\beta_{Cos}$ | −0.019 (−0.91) | | | $\beta_{Cos}$ | −0.008 (−0.49) | | |
| $\beta_{BM2}$ | | −0.023 (−0.96) | | $\beta_{BM2}$ | | −0.014 (−0.68) | |
| $\beta_{BHS}$ | | | −0.025 (−1.23) | $\beta_{BHS}$ | | | −0.015 (−0.74) |
| Macroeconomic Variables | YES | YES | YES | Macroeconomic Variables | YES | YES | YES |
| $Adj.R^2$ | 0.48 | 0.50 | 0.51 | $Adj.R^2$ | 0.40 | 0.43 | 0.44 |

*Note:* The significance levels are adjusted using the Newey-West method, with the lag length selected as specified $T^{0.25} \approx 4$. ***, ** and * denote significance at the 1%, 5%, and 10% levels, respectively. The values in parentheses are Newey-West adjusted $t$ values.

reveal the asymmetric pricing mechanism of the market toward extreme interest rate movements, and enrich the nonlinear theory of the interest rate term structure.

We use China's overnight Shanghai Interbank Offered Rate (Shibor) and the 10-year Treasury bond yield as proxies for short-term and long-term interest rates, respectively. The main findings of our study can be summarized as follows: The systematic skewness factors demonstrate significant predictive power for future changes in both China's overnight Shibor and the 10-year Treasury bond yield, with a negative correlation observed. This indicates that investors' current hedging demands affect future interest rates by influencing future monetary policy formulation. When investors' hedging demands rise, they increase purchases of stocks with positive systematic skewness, leading to the overvaluation of these stocks, lower future returns, and an increase in the current value of the systematic skewness factors. This signal is transmitted through monetary policy and significantly impacts future interest rates.

Based on this, we performed robustness checks via endogeneity tests, subsample analysis, out-of-sample forecasts, and comparisons with benchmark models. The results confirm that systematic skewness factors have significant predictive power for future interest rates. Moreover, for both short-term and long-term rates, prediction errors based on systematic skewness factors are consistently lower than those from an AR model and the extended Taylor-rule model proposed by Ma et al. (2025) [12]. Heterogeneity analysis further shows that in bull markets, systematic skewness factors forecast short-term rates more accurately over the near term than long-term rates, whereas in bear markets, their predictive power for long-term rates persists longer than for short-term rates. This pattern reflects differences in investor behavior across market cycles.

In summary, systematic skewness can serve as an asymmetric pricing signal in the market for extreme interest rate risks. Its increase often indicates a rise in investors' anxiety over liquidity tightening, thereby providing central banks with a forward-looking sentiment monitoring window independent of traditional economic indicators. Our empirical design and findings contribute to the existing research mainly in the following aspects: (1) From the perspective of investors' hedging demands on predicting future interest rates, it provides a macroeconomic explanation for the formation of the systematic skewness pricing effect. (2) It incorporates higher-moment risk pricing into the interest rate analysis framework, reveals the asymmetric pricing mechanism of the market toward extreme interest rate movements, and enriches the nonlinear theory of the interest rate term structure. (3) It offers new evidence on monetary policy transmission.

## Supporting information

**S1 File. Data1.zip.** Data1.zip comprises data such as the monthly price change of the SSE index, dividend yield, daily stock data and daily price change of the SSE index.
(ZIP)

**S2 File. Data2.zip.** Data2.zip contains data including trading date, total owner's equity, total asset growth rate, risk-free rate, return on equity and monthly stock data.
(ZIP)

## Acknowledgments

The authors would like to thank the editor and the anonymous reviewers for their valuable comments and suggestions, which have greatly improved the quality of this paper.

## Author contributions

**Conceptualization:** Xinyao Liang.

**Data curation:** Yichen Sun.

**Formal analysis:** Xinyao Liang.

**Funding acquisition:** Xinyao Liang.

**Investigation:** Xinyao Liang, Yichen Sun.

**Methodology:** Xinyao Liang.

**Project administration:** Xinyao Liang.

**Resources:** Xinyao Liang.

**Software:** Xinyao Liang.

**Supervision:** Xinyao Liang.

**Validation:** Xinyao Liang, Yichen Sun.

**Visualization:** Xinyao Liang.

**Writing – original draft:** Xinyao Liang.

**Writing – review & editing:** Xinyao Liang.

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
