## [Decision Letter · Decision Letter 0]

5 Nov 2025

Dear Dr. Liang,

Thank you for submitting your manuscript to PLOS ONE. After careful consideration, we feel that it has merit but does not fully meet PLOS ONE’s publication criteria as it currently stands. Therefore, we invite you to submit a revised version of the manuscript that addresses the points raised during the review process.

We look forward to receiving your revised manuscript.

Kind regards,

Ricky Chee Jiun Chia

Academic Editor

PLOS ONE

**Journal Requirements:**

1. When submitting your revision, we need you to address these additional requirements. Please ensure that your manuscript meets PLOS ONE's style requirements, including those for file naming. The PLOS ONE style templates can be found at https://journals.plos.org/plosone/s/file?id=wjVg/PLOSOne_formatting_sample_main_body.pdf and https://journals.plos.org/plosone/s/file?id=ba62/PLOSOne_formatting_sample_title_authors_affiliations.pdf 2. Thank you for stating in your Funding Statement: This study is supported by the Natural Science Foundation of Sichuan Province (2025NSFSC1959).  Please provide an amended statement that declares *all* the funding or sources of support (whether external or internal to your organization) received during this study, as detailed online in our guide for authors at http://journals.plos.org/plosone/s/submit-now.  Please also include the statement “There was no additional external funding received for this study.” in your updated Funding Statement. Please include your amended Funding Statement within your cover letter. We will change the online submission form on your behalf. 3. We note that there is identifying data in the Supporting Information files. Due to the inclusion of these potentially identifying data, we have removed this file from your file inventory. Prior to sharing human research participant data, authors should consult with an ethics committee to ensure data are shared in accordance with participant consent and all applicable local laws. Data sharing should never compromise participant privacy. It is therefore not appropriate to publicly share personally identifiable data on human research participants. The following are examples of data that should not be shared: -Name, initials, physical address-Ages more specific than whole numbers-Internet protocol (IP) address-Specific dates (birth dates, death dates, examination dates, etc.)-Contact information such as phone number or email address-Location data-ID numbers that seem specific (long numbers, include initials, titled “Hospital ID”) rather than random (small numbers in numerical order) Data that are not directly identifying may also be inappropriate to share, as in combination they can become identifying. For example, data collected from a small group of participants, vulnerable populations, or private groups should not be shared if they involve indirect identifiers (such as sex, ethnicity, location, etc.) that may risk the identification of study participants. Additional guidance on preparing raw data for publication can be found in our Data Policy (https://journals.plos.org/plosone/s/data-availability#loc-human-research-participant-data-and-other-sensitive-data) and in the following article: http://www.bmj.com/content/340/bmj.c181.long. Please remove or anonymize all personal information (<specific identifying information in file to be removed>), ensure that the data shared are in accordance with participant consent, and re-upload a fully anonymized data set. Please note that spreadsheet columns with personal information must be removed and not hidden as all hidden columns will appear in the published file. 4. Please include captions for your Supporting Information files at the end of your manuscript, and update any in-text citations to match accordingly. Please see our Supporting Information guidelines for more information: http://journals.plos.org/plosone/s/supporting-information. 5. If the reviewer comments include a recommendation to cite specific previously published works, please review and evaluate these publications to determine whether they are relevant and should be cited. There is no requirement to cite these works unless the editor has indicated otherwise. 

Reviewers' comments:

**Comments to the Author**

1. Is the manuscript technically sound, and do the data support the conclusions?

Reviewer #1: Yes

Reviewer #2: Yes

2. Has the statistical analysis been performed appropriately and rigorously?

Reviewer #1: Yes

Reviewer #2: Yes

3. Have the authors made all data underlying the findings in their manuscript fully available?

Reviewer #1: Yes

Reviewer #2: No

4. Is the manuscript presented in an intelligible fashion and written in standard English?

Reviewer #1: Yes

Reviewer #2: Yes

**Reviewer #1:**  The manuscript provides an original empirical contribution at the intersection of asset-pricing theories and macro-finance. It convincingly connects systematic skewness preferences of investors with future interest rate changes, offering a macroeconomic rationale for pricing effects related to systematic skewness.

a) Original conceptual framing - the authors combine behavioural-finance intuition with macroeconomic forecasting.

b) Rigorous empirical strategy - the authors use three independent measures of skewness (Cos, BM2, BHS).

c) Control variables - consider a extensive list of control variables, on the macroeconomic side, we have M2, GDP, CPI, loans and US rates; in addition, the authors included a complete set of classic Fama–French factors.

d) Heterogeneity aspects - the authors analyse bull and bear market effects, which creates considerable interpretive richness.

e) Statistical attributes - Methods such as Newey–West adjustments are appropriate, all specifications suggest a similarly

Positive Features.

1. Theoretical aspect: Offer a clearer equilibrium mechanism related to systematic skewness and interest-rate changes, considering shedding light on this through a simple model or cost of risk-hedging mechanisms.

2. Causality implication: the reverse-causality issue impacts practical use and should be presented or examined. One strategy is to offer instrumental-variable estimation or some method of Granger-causality to asses the direction, whether intentional or implicit.

3. Economic significance: Quantify improvement in forecasting ability to be able to weigh against a baseline state-space model (for example, AR, VAR, Taylor rule).

4. Improvements from sample results are good and need some discussion regarding potential bias from investors' preference towards small-cap stocks' exclusion. Identify sub-period analysis as a means to capture changing monetary regimes within China.

5. Better presentation-in terms of congested and extensive tables; perhaps add a schematic from the author's insights for the reader to absorb the conceptual relationships as the state of the literature creates large gaps.

6. Policy relevance: In further discussing and offering text of implications from monetary-policy signalling, portfolio management, and/or in monitoring market-based expectations.

The paper presents strong methodological aspects, as well as, strong indications that the paper holds theoretical potential as it relates to behavioural-finance applications in both decision-making and policy. The pdf merits consideration for publication status upon noticeable revisions. The paper should make a suitable learning contribution to scholarly articles and apply useful theories to finance. Discovered relationships refining the forecasting and practical implications aim to be substantive through careful revision.

**Reviewer #2:**  This study meaningfully advances asset pricing research by integrating systematic skewness with macroeconomic expectations theory. It contributes both empirically and conceptually, offering a nuanced explanation for skewness-based pricing effects that goes beyond behavioural narratives. Future refinements focusing on causal inference and broader cross-market comparisons would elevate its impact further. The analysis incorporates nearly 3,700 stocks over a 25-year span, employs rigorous data filters, and conducts separate investigations for bull and bear market cycles, offering insights into behavioural heterogeneity across regimes

1. the study establishes correlation between systematic skewness factors and future interest rates but should further address potential endogeneity or reverse causality

2. Detailed explanations of regression frameworks, control variables, and robustness checks should be provided

3. It would be valuable to discuss whether the results hold across different sample periods, especially considering China’s structural monetary shifts and policy-driven market influences.

4. Regression outputs are well-detailed, but further clarity regarding the selection/lag structure of control variables and additional robustness checks could improve reader confidence.

5. While the theoretical rationale behind investor expectations is plausible, it may benefit from linking to specific macro indicators (e.g., inflation expectations, credit growth) that could mediate the skewness-interest rate relationship.

6. the paper interprets prolonged predictive effects in bear versus bull markets through investor behaviour. Deeper engagement with alternative channels (e.g., policy, liquidity, foreign capital flows) would round out the narrative.

7. Benchmarking results with evidence from developed markets could strengthen the argument for the uniqueness or generalizability of these findings.

**Do you want your identity to be public for this peer review?** For information about this choice, including consent withdrawal, please see our Privacy Policy

Reviewer #1: No

Reviewer #2: **Yes:** Dr. Kaustav Aditya, Senior Scientist, ICAR-IASRI, New Delhi

---

## [Author Response · Author response to Decision Letter 1]

23 Dec 2025

Dear Editor,

We sincerely appreciate your and all the reviewers’ critiques and valuable feedback. We are fortunate that our manuscript was eventually assigned to these reviewers, as their valuable feedback not only helped us improve the paper but also provided some insightful directions for future research. Please convey our heartfelt thanks to these experts.

Based on the feedback received, we have carefully revised the original manuscript. All changes are highlighted in blue text within the revised version. Below, you will find our point-by-point responses to the comments. (At the same time, we have uploaded an attachment titled "Response to Reviewers".)

We hope the new version meets your standards, and if any shortcomings remain, please do not hesitate to point them out. We look forward to receiving any response regarding our submission and are pleased to address any further questions or comments you may have.

Xinyao Liang

Yichen Sun

(1) Your Comment:

Theoretical Development: While the empirical findings are robust, the theoretical mechanism remains underdeveloped. The paper would benefit from a formal model or more rigorous theoretical framework explaining why systematic skewness factors should predict interest rates beyond the intuitive narrative. What is the equilibrium relationship? How do rational expectations reconcile with the documented patterns?

Our revision:

Thank you for this suggestion for revision. We completely agree with your perspective.

Investors’ hedging activities may trigger market liquidity tensions or asset sell-offs, increasing the vulnerability of the financial system. If such behavior exacerbates systemic risks, the central bank may adjust interest rates (for example, by lowering rates to provide liquidity) to smooth market volatility and maintain financial stability (Drechsler et al., 2021; Bocola and Lorenzoni, 2020).

Stocks exhibiting positive systematic skewness, which demonstrate the ability to generate positive returns during market downturns and thus meet investors’ hedging demands (Langlois, 2020; Baltussen et al., 2021), tend to be favored by investors when their hedging demands increase. This preference drives up the prices of these stocks, leading to lower future returns and consequently enhancing the value of the systematic skewness factor. Conversely, when investors’ hedging demands decline, the value of the systematic skewness factor also diminishes. Based on the above reasoning, we define the systematic skewness factors as the return difference between the 30% of stocks with the lowest (typically negative) systematic skewness and the 30% of stocks with the highest (typically positive) systematic skewness, and we employ numerical changes of systematic skewness factors to capture the time-varying nature of investors’ hedging demands.

To further elaborate on the theory, we construct a two-period model to illustrate the relationship between investors’ expectations of future interest rates and their hedging demands. This content has been added in Section 3 of the revised manuscript.

(2) Your Comment:

Endogeneity and Causality: The most significant methodological concern is potential reverse causality. Do systematic skewness factors predict interest rates, or do expectations about interest rates drive systematic skewness preferences? The paper treats predictive regressions as sufficient evidence, but instrumental variable approaches, Granger causality tests, or natural experiments would substantially strengthen causal claims. Current evidence demonstrates correlation, not causation.

Our revision:

Thank you for this suggestion for revision. We completely agree with your perspective.

In Section 5.1 of the revised manuscript, we have supplemented Granger causality tests and instrumental variable tests to strengthen causal inference.

(3) Your Comment:

Economic Magnitude: The paper reports statistical significance extensively but provides limited discussion of economic significance. What is the practical forecasting improvement over benchmark models? How do the predictive R2 values compare to standard interest rate forecasting models? A comparison with AR models, Taylor rule specifications, or term structure models would contextualize the findings.

Our revision:

Thank you for this suggestion for revision. We completely agree with your perspective.

In Section 5.3 of the revised manuscript, we have added out-of-sample forecasting and comparisons with benchmark models.

(4) Your Comment:

Sample Selection and Data Concerns: Excluding the smallest 30% of stocks by market capitalization to avoid “shell value” effects is pragmatic but potentially problematic. This could induce selection bias and limit generalizability. Additionally, the paper’s treatment of the 1997-2024 sample period deserves scrutiny - this encompasses multiple regime changes in Chinese monetary policy and market regulation that warrant explicit discussion or subsample analysis.

Our revision:

Thank you for this suggestion for revision. We completely agree with your perspective.

In the revised manuscript, we have adjusted the data sample by no longer excluding the smallest 30% of stocks by market capitalization and have re-reported all empirical results accordingly. These modifications are reflected throughout the empirical findings and are highlighted in blue text.

Additionally, in Section 5.2, we divided the sample into bull-market and bear-market phases and separately reported the empirical results for these two subsample tests. Moreover, we also used the 2005 equity division reform in Chinese stock market as a breakpoint, dividing the sample into pre-2005 and post-2005 periods. The empirical results similarly support the research conclusions of this paper. However, given the already substantial length of the manuscript, these findings were not included in the revised version. If required, we would be glad to supplement this part of the empirical results into the revised manuscript.

(5) Your Comment:

Robustness Checks: The paper lacks several standard robustness tests including alternative forecasting horizons beyond 1-4 months, out-of-sample forecasting performance, and alternative specifications for constructing systematic skewness factors (e.g., different portfolio cutoffs than 30/30).

Our revision:

Thank you for this suggestion for revision. We completely agree with your perspective.

In the revised manuscript, we report the predictive ability of the systematic skewness factor for interest rates up to four months ahead across all empirical tables. For forecasting horizons beyond four months, the systematic skewness factor does not demonstrate significant predictive power for future interest rates, and thus such results are not included in the revision.

Additionally, in Sections 5.3 and 5.4, we present out-of-sample forecasting and alternative specifications for constructing systematic skewness factors, respectively.

(6) Your Comment:

Readability and Presentation: While the paper is well-organized, several tables (especially Tables 3~6) are excessively dense. The addition of graphical summaries, conceptual diagrams, or flowcharts illustrating relationships between skewness, market sentiment, and interest rates would enhance readability and accessibility for a multidisciplinary audience.

Our revision:

Thank you for this suggestion for revision. We completely agree with your perspective.

In the revised manuscript, we have modified the relevant tables and added Figure 1 to illustrate the relationships among skewness, market sentiment, and interest rates.

(7) Your Comment:

Policy and Practical Implications: The conclusion section could more explicitly articulate the implications for monetary policy, portfolio management, and risk assessment. For instance, how might systematic skewness serve as a market-based indicator of future policy shifts or investor sentiment during liquidity tightening cycles?

Our revision:

Thank you for this suggestion for revision. We completely agree with your perspective.

In the revised manuscript, we have emphasized in multiple sections that when investors’ hedging demand increases, they tend to purchase more stocks with positive systematic skewness, leading to the overvaluation of these stocks, a decline in future returns, and a corresponding rise in the value of the systematic skewness factor. This signal is transmitted through monetary policy and significantly influences future interest rates. Moreover, for both short-term and long-term interest rates, prediction errors based on systematic skewness factors are consistently lower than those from an AR model and the extended Taylor‑rule model proposed by Ma et al. (2025). Systematic skewness can serve as an asymmetric pricing signal in the market for extreme interest rate risks. Its increase often reflects heightened investor anxiety over liquidity tightening, thereby offering central banks a forward-looking sentiment monitoring window independent of traditional economic indicators.

---

## [Editor Report · Decision Letter 1]

11 Jan 2026

Can Systematic Skewness Factors Predict Future Interest Rates: Evidence from China

PONE-D-25-52488R1

Dear Dr. Xinyao Liang,

We’re pleased to inform you that your manuscript has been judged scientifically suitable for publication and will be formally accepted for publication once it meets all outstanding technical requirements.

Kind regards,

Ricky Chee Jiun Chia

Academic Editor

PLOS One
---

## [Editor Report · Acceptance letter]

PONE-D-25-52488R1

PLOS One

Dear Dr. Liang,

I'm pleased to inform you that your manuscript has been deemed suitable for publication in PLOS One. Congratulations! Your manuscript is now being handed over to our production team.

Kind regards,

on behalf of

Dr. Ricky Chee Jiun Chia

Academic Editor

PLOS One